# Is overwintering mortality driving enigmatic declines? Evaluating the impacts of trematodes and the amphibian chytrid fungus on an anuran from hatching through overwintering

Olivia Wetsch *◉, Miranda Strasburg◉◉, Jessica McQuigg, Michelle D. Boone

Department of Biology, Miami University, Oxford, Ohio, United States of America

◉ These authors contributed equally to this work.
* wetschoe@miamioh.edu

**Data Availability Statement:** All relevant data are available from the Figshare database (URL: https://figshare.com/s/9711ca7805cd5d72cedb).

## Abstract

Emerging infectious diseases are increasing globally and are an additional challenge to species dealing with native parasites and pathogens. Therefore, understanding the combined effects of infectious agents on hosts is important for species' conservation and population management. Amphibians are hosts to many parasites and pathogens, including endemic trematode flatworms (e.g., *Echinostoma* spp.) and the novel pathogenic amphibian chytrid fungus (*Batrachochytrium dendrobatidis* [*Bd*]). Our study examined how exposure to trematodes during larval development influenced the consequences of *Bd* pathogen exposure through critical life events. We found that prior exposure to trematode parasites negatively impacted metamorphosis but did not influence the effect of *Bd* infection on terrestrial growth and survival. *Bd* infection alone, however, resulted in significant mortality during overwintering—an annual occurrence for most temperate amphibians. The results of our study indicated overwintering mortality from *Bd* could provide an explanation for enigmatic declines and highlights the importance of examining the long-term consequences of novel parasite exposure.

## Introduction

Current trends show that disease-related wildlife declines are occurring at unprecedented rates and that they are spurred by human modification of landscapes [1, 2]. The rise of emerging infectious diseases poses an additional challenge to species already managing natural parasite and pathogen infections [3, 4]. Understanding how hosts manage co-infections of either native and/or novel parasites and pathogens is of utmost importance to humans and wildlife given that co-infection is the norm rather than the exception [5].

Amphibians are declining globally at a concerning rate [6], faster than any other vertebrate group [7], due in part to emerging infectious agents [8–10]. Amphibians may be particularly

**Funding:** All authors and experimental materials were funded by Miami University Department of Biology. The funders had no role in study design, data collection and analysis, decision to publish, or preparation of the manuscript.

**Competing interests:** The authors have declared that no competing interests exist.

susceptible to novel diseases because of vulnerable periods at metamorphosis and during over-wintering. Amphibian metamorphosis involves a dramatic restructuring of the immune system [11], which can lead to immunosuppression, thereby potentially increasing vulnerability to infectious agents [12]. Overwintering of temperate and subarctic species is a physiologically stressful period that can also reduce immune function [13], allowing otherwise benign infections to cause mortality not anticipated from effects prior to overwintering [14, 15]. Given that disease-driven mortality may be difficult to detect during metamorphosis and overwintering when death is not easily observed in the field, it is critical to evaluate the impacts of common infectious agents like the amphibian chytrid fungus (*Batrachochytrium dendrobatidis*, *Bd*) and trematodes (e.g., *Echinostoma* spp.) across multiple life stages and through critical life events.

Amphibian chytridiomycosis, caused by the fungal pathogen *Batrachochytrium dendrobatidis* (*Bd*), is one of the most devastating wildlife diseases in known history [16, 17]. *Bd* encysts in keratinized skin and/or jaw sheaths of amphibians and has been found to reduce growth and body condition [18–20], impact time to metamorphosis [21], and affect survival prior to and through overwintering [15, 18, 22]. Amphibians could be more susceptible to this novel infectious disease-agent based on body condition or exposure to other parasites that individuals encounter in the natural environment.

Trematode parasites are globally important for human and wildlife health. In amphibians, one trematode genus, *Echinostoma*, establishes in kidneys during the larval stage and can cause mortality in some cases such as when exposure occurs early in development or when infection loads are high [23, 24]. In most cases, however, infections by *Echinostoma* spp. (hereafter "trematodes") are not fatal. Instead, trematodes often have sublethal effects, such as developmental delays [25, 26] and increased movement to avoid infection [27, 28], which could increase their susceptibility to other parasites or pathogens by reducing their body condition. Though native to North America, these trematodes may be increasing in abundance from nutrient inputs, which increase food resource availability for their intermediate snail hosts [24] and which may influence disease dynamics.

The objectives of this study were to 1) determine the individual and interactive effects of larval exposure to trematodes and exposure to *Bd* at metamorphosis on Blanchard's cricket frogs (*Acris blanchardi*) reared from hatching through terrestrial overwintering; 2) evaluate how exposure to one parasite, *Echinostoma* trematodes, during larval development influences subsequent susceptibility to the *Bd* pathogen; and 3) use a population growth model to assess population-level implications of trematodes and *Bd*. Our study species, the Blanchard's cricket frog, is experiencing enigmatic declines throughout portions of its range and is a common host for trematodes and *Bd* [29, 30]. Therefore, understanding how these infectious agents interact to influence this declining species is of paramount importance. We hypothesized that 1) infection by trematodes will increase susceptibility to *Bd*, and 2) that exposure to single or multiple infectious agents will increase cricket frog mortality at metamorphosis and during overwintering in the terrestrial environment.

## Materials and methods

### Ethics statement

All aspects of our study design and protocol were reviewed and approved by the office of Research Ethics and Integrity Program at Miami University through the Institutional Animal Care and Use Committee (IACUC) via protocol IACUC 827. Animal collection was approved by Ohio Department of Natural Resources (Wild Animal Permit: Scientific Collection #20–177). Individuals that survived larval rearing but were not used for terrestrial rearing (totaling 50 individuals) and individuals that survived overwintering (totaling 29 individuals) were

immediately and humanely euthanized using 1% buffered MS-222 as required by our Ohio Department of Natural Resources collecting permit. Mortality during larval rearing under semi-natural field conditions totaled 82 tadpoles and two juveniles died during terrestrial rearing prior to overwintering. Mortality during overwintering totaled 37 individuals and is presumably disease-driven as was expected per the study design. All participants of the study received CITI animal care training provided by Miami University.

## Animal collection

We collected six amplexed Blanchard's cricket frog pairs on 9-Jun-2017 from a private pond in Oxford, Ohio (Butler County; 39˚31'29.6" N, 84˚44'25.8" W) and allowed pairs to oviposit in the laboratory overnight in plastic shoebox containers with water from the originating pond. We held the clutches in the laboratory between 23–25˚C until they were added to outdoor mesocosms on 28-Jun. In the laboratory, we fed the tadpoles ground fish flakes (Tetra Holdings) *ad libitum* and changed water daily.

We collected ramshorn snails (*Helisoma trivolvis*) by hand from two local ponds that differed in their infection status of trematodes. We collected infected snails from a pond at Miami University's Ecology Research Center (ERC; Oxford, OH; 39˚31'42.7" N, 84˚43'24.9" W) on 29-Jun, and uninfected snails from Bachelor Pond in Miami University's Natural Areas (39˚31'18.5" N, 84˚42'26.4" W) on 30- Jun. *Echinostoma* spp. trematodes require three hosts—two intermediate hosts and one definitive host; within the definitive host (a bird or mammal), the adult trematode undergoes sexual reproduction releasing eggs into the host's feces [22]. When the trematode eggs reach the water, they hatch into miracidia, free-swimming larvae that infect aquatic snails. Within the snail host, these parasites develop into rediae, which undergo asexual reproduction to produce cercariae. Free-swimming cercariae leave their snail host and form metacercaria in the amphibian host, which must be consumed by the definitive host for the parasite to complete its life cycle. At the time of experimental setup snails were not actively shedding cercariae, so we dissected a subset of 20 snails from each pond to determine infection prevalence. These dissections revealed 50% of the dissected snails from the ERC pond contained rediae and developing cercariae identified as *Echinostoma* spp. using Schell [31], and no snails were infected from Bachelor Pond.

## Trematode exposure in simulated aquatic communities

In our study, we reared cricket frogs from hatching through metamorphosis during which time they were exposed to the presence or absence of trematode parasites via snails; following metamorphosis, cricket frogs were exposed to *Bd* in the laboratory to examine how early life trematode exposure influenced susceptibility to *Bd*. To summarize our experimental design, we had two larval trematode treatments (present or absent) and two terrestrial *Bd* treatments (present or absent) for four treatments total.

To initiate our study, we first created artificial communities in 10 polyethylene mesocosms (1.85 m in diameter, 1,480 L volume). We added 1000 L of city water (9-Jun), 1 kg of well-mixed leaf litter collected from a deciduous forest in Miami University's Natural Areas (10-Jun), and inoculations of zooplankton and algae to mesocosm water from the ERC pond (12-16-June). We covered each mesocosm with a screen lid to prevent the colonization of non-target species.

On 28-Jun, we randomly assigned 20 free-swimming (Gosner 25 [32]) tadpoles to each mesocosm. To manipulate exposure to trematodes, on 30-Jun-2017 (experimental day 0) we randomly assigned either 25 snails (diameter: 12.76 mm ± 1.88 [mean ±1 SD]) from the ERC Pond (that were expected to have ~50% infection prevalence) or 25 snails from Bachelor Pond

(that were expected to be uninfected) to each mesocosm and replicated each trematode treatment five times (2 trematode treatments [present or absent] x 5 replicates = 10 mesocosms). We added snails to all mesocosms to control for effects of competition between snails and tadpoles for food resources. The snail density is within the range of natural densities (range 0 to 1,684 planorbid snails per m$^2$; average 587 snails per m$^2$ [33]) and ensured exposure to trematodes in trematode-present mesocosms.

We monitored mesocosms daily and removed any individuals reaching metamorphosis (presence of at least one front limb; Gosner 42 [32]). Once each metamorph reached Gosner stage 46 ($\leq$3 days), we weighed them to the nearest milligram, determined time and survival to metamorphosis, and transferred them into terrestrial terraria (detailed below). We terminated the mesocosm portion of the experiment on 14-Sep (experimental day 77). We drained the mesocosms and searched through the leaf litter for remaining tadpoles ($\leq$3 tadpoles/mesocosm).

## Rearing in terrestrial laboratory environments and *Bd* exposure

We randomly assigned metamorphs from each larval trematode treatment to a *Bd* treatment (present or absent) and reared individuals in terraria through overwintering (described below). We had unequal sample sizes due to higher mortality from larval trematode exposure, which resulted in use of every metamorph collected from trematode-exposed ponds; one metamorph collected from a trematode exposure pond was dissected and it was infected with trematode parasites. Consequently, our experiment consisted of 19 replicates in the control (no trematode and no *Bd* exposure), 15 for trematode only exposure, 20 for *Bd* only exposure, and 14 for trematode plus *Bd* exposure, which resulted in 68 individuals reared post-metamorphosis.

We transferred individuals at metamorphosis into 2 L beakers lined with ~5 cm layers of pea gravel and moist topsoil, and we covered with a fiberglass mesh screen to prevent escape. In each terrarium, we included a 60 mm petri dish with water. We checked survival daily and fed all individuals three times per week with 3.2 mm crickets that were approximately 10% of the average body weight of juveniles until overwintering simulation began. During each feeding period, we refilled the petri dish and moistened the soil. We held the terraria in a temperature-controlled environmental chamber on a 14h:10h light:dark cycle at 23˚C. We weighed all individuals every two weeks until overwintering was simulated.

We held juvenile frogs in the laboratory between 0–21 days before *Bd* exposure on 31-Aug (terrestrial rearing day 0), to ensure sufficient sample sizes for each treatment. We cultured *Bd* isolate JSOH01 (Toledo, OH, USA; isolated from *Rana pipiens*) using standardized procedures [34] beginning 8-Aug. To manipulate *Bd* exposure, we placed all frogs in ventilated plastic petri dishes for 12 h containing 7 mL of dechlorinated water (the frog's entire ventral surface was submerged in the water), along with 1 mL of either *Bd* or a *Bd*-free agar plate wash [35]. For the agar plate wash, we added dechlorinated water to either sterile agar plates or plates containing *Bd* zoospores for 30 minutes and decanted the solution to create an exposure solution. For the *Bd* treatment, we exposed individuals with 2.007 x 10$^6$ cells.

Four weeks post-*Bd* exposure on 2-Oct (terrestrial rearing day 33), we swabbed all frogs for *Bd* using a standardized swabbing technique of five passes over each of the ventral abdominal surface, rear limbs, and rear feet. We again swabbed individuals before overwintering on terrestrial rearing day 74 and post overwintering on terrestrial rearing day 182. Swabs were stored at -20˚ C until analysis. We extracted DNA from cotton-tipped wooden swabs collected 4 weeks post *Bd* exposure and post overwintering using a protocol adapted from Boyle et al. [36]; pre-overwintering swabs were not analyzed because *Bd*-exposed animals surviving

overwintering remained infected. Adaptations to protocol included the use of spin baskets to collect DNA supernatant. These spin baskets were constructed by suspending a sterile 0.7 mL microcentrifuge tube with a small hole (14-gauge needle) containing a *Bd* swab within a sterile 1.5 mL microcentrifuge tube. This design ensured that the liquid containing DNA could be harvested from each swab after spinning as the liquid was pulled through the hole, but the swab was retained within the inner 0.7 mL tube. We pooled 5 uL of each control sample to make a composite sample for each non-*Bd* treatment. We diluted all samples to 2ng/μl DNA with RNAse-free water to overcome inhibition as a result of the wooden swabs and analyzed all samples in triplicate using qPCR [36]. Dilution of DNA samples to overcome inhibition associated with organically derived compounds is a common practice yet can cause a reduction in qPCR precision [37]. As such, we could not quantify *Bd* load due to poor precision but confirmed presence of *Bd* if at least two out of three wells were positive per swab. If one out of three wells was positive, we reran the sample and considered it positive it at least two out of three wells was positive.

Ten weeks post-*Bd* exposure, we simulated overwintering conditions following adapted methods of James [38]. Beginning on terrestrial rearing day 74, we decreased the rearing temperature and number of crickets fed gradually until 17˚C was reached on terrestrial rearing day 78. On terrestrial rearing day 74, we also stocked terraria with 2.5 cm of additional soil and leaf litter to reduce terraria desiccation. We held the frogs at 17˚C until terrestrial rearing day 88 (10 days) at a 10:14 light:dark cycle to match local conditions, and to allow for gut clearance to prevent intestinal infections during overwintering. On terrestrial rearing day 88, we dropped the temperature to 7˚C, and then incrementally decreased the temperature to 3˚C where it was held until terrestrial rearing day 182. We maintained moisture in each terrarium by spraying the leaf litter with dechlorinated water weekly. On terrestrial rearing day 182, we removed all frogs from the overwintering chamber and brought them to room temperature. We weighed all surviving frogs, swabbed them for *Bd*, and then euthanized them in 1% buffered MS-222. We preserved euthanized frogs in 10% neutral buffered formalin, which was replaced with 75% ethanol after 2 days. All dead individuals were recorded. To quantify the number of metacercarial trematode cysts, we dissected all preserved frogs and removed both kidneys. We then flattened the kidneys between two glass slides to expose individual metacercarial cysts and examined each slide under a dissecting microscope.

## Statistical analysis

For the mesocosm portion of our study, mesocosm was used as the experimental unit. We used a generalized linear model (GLM) with a binomial distribution to test for the effects of trematode exposure on survival to metamorphosis. We used ANOVA to test for effects of trematodes on time to metamorphosis and size at metamorphosis.

For the terrestrial portion of the experiment, the experimental unit was the individual because individuals were reared alone. We used repeated-measures ANOVA to examine the effects of trematode exposure, *Bd* infection, and their interaction on mass of individuals over time. We tested for the effects of trematode exposure, *Bd* infection, and their interaction on terrestrial survival and overwintering survival of individuals using logistic regression. We also tested for the effect of individual mass prior to overwintering on overwintering survival using logistic regression.

To determine if trematodes or *Bd* influenced overwintering survival through effects on terrestrial mass, we performed a path analysis using the 'piecewiseSEM' package in R [39]. The model evaluated if the indirect effects of trematode exposure or *Bd* infection (coded as 0 or 1 using a linear model) on mass prior to overwintering influenced survival through

overwintering (coded as 0 or 1 using a general linear model with a binomial distribution). We evaluated overall path model fit with Shipley's test of d-separation, which calculates a Fisher's C test and compares it to a Chi-squared distribution where p > 0.05 indicates a good fitting model [40]. If a missing path was detected with the d-separation test, we added that path.

## Population growth model

To determine if the effects of trematodes and *Bd* on survival could impact population growth in Blanchard's cricket frog populations, we developed a stage-based population model to estimate lambda (λ). We modeled population growth under four conditions: no exposure to trematodes or *Bd*, exposure to trematodes, exposure to *Bd*, and exposure to both trematodes and *Bd*. To understand the effect of trematodes and *Bd*, we reduced pre-metamorphic survival by 30% and juvenile survival of juveniles by 88%, respectively, based on results from the present study.

Cricket frogs reach sexual maturity within one year [41] and seldom survive for a second year of breeding [42]; therefore, cricket frogs rarely have overlapping stages, so our model differed from the stage-based projections matrix models commonly used in the literature to model anuran population growth (i.e., [15, 43, 44]; Fig 1). Instead, to determine population growth rate, we multiplied estimates of survival at two different life stage transitions—pre-metamorphic (embryo and larval) and juvenile—by the birth rate (i.e., fecundity [45]).

We used values from the literature to determine vital rates in our model (Table 1). Field measures of embryo and larval survival are rare in the literature and were not available for Blanchard's cricket frogs, so we used range of pre-metamorphic survival from wood frogs (*Rana sylvatica*) determined by Berven's [46] study monitoring wood frog populations over 7 years. It is common in demographic transition models when species specific vital rates are unknown to use rates associated with similar species [15, 43]. We used survival rates of overwintered Blanchard's cricket frog metamorphs reared in outdoor terrestrial enclosure as mean juvenile survival rate [47]. We used the assumption of Biek et al. [43] for probability of laying a clutch (mean 1, SD 0); because only females lay clutch, we divided clutch size by 2. Clutch size

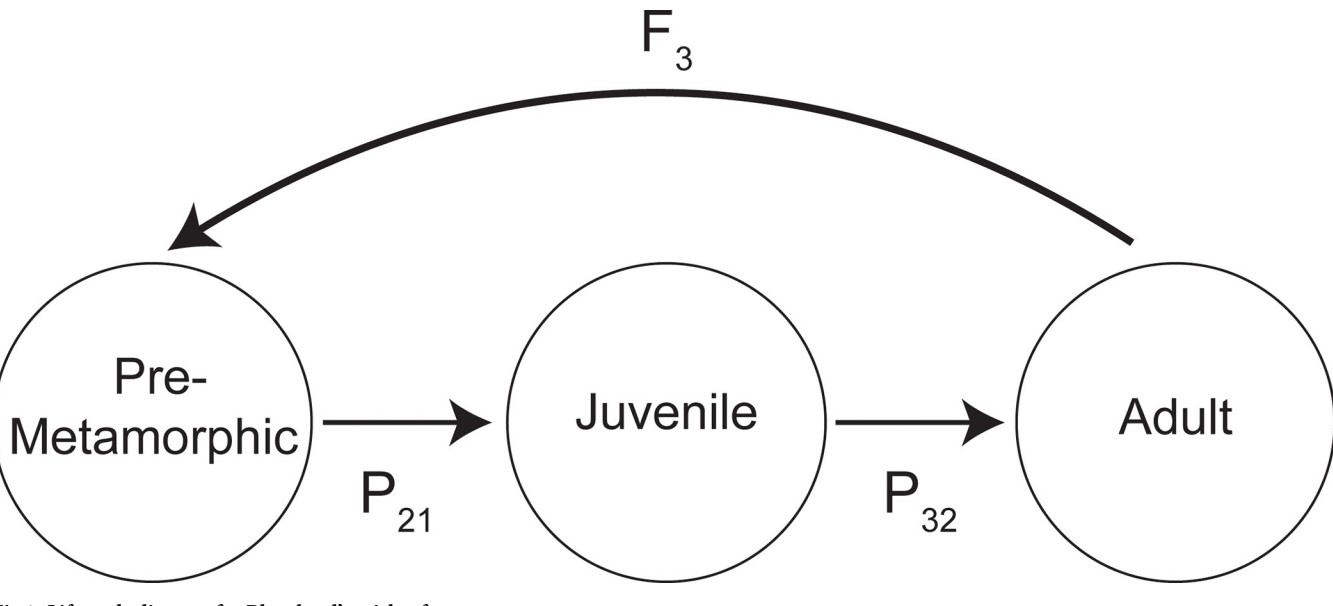

**Fig 1. Life-cycle diagram for Blanchard's cricket frogs.**

**Table 1. Vital rates and transition probabilities used in the stage-structure model.**

| Vital rate | Mean (SD) | Species |
|---|---|---|
| Pre-metamorphic survival | Range: 0.01 to 0.08 [46] | *Rana sylvatica* |
| Juvenile survival | Mean (SD): 0.34 (0.03) [47] | *Acris blanchardi* |
| Clutch size | Mean (SD): 266 (64) [48] | *A. blanchardi* |
| Probability of laying | 1 [15, 43] | *R. aurora, R. temporaria,* and *R. pipiens* |

was based on mean estimates from Trauth et al. [48] for Blanchard's cricket frogs. We inferred standard deviation of juvenile survival and clutch size from reported ranges as in Biek et al. [43].

To account for demographic stochasticity, we calculated the mean finite rate of increase (λ) for Blanchard's cricket frog populations for each of the treatments after 2000 iterations where vital rates were randomly selected from a distribution. We used a uniform distribution with the range for pre-metamorphic survival, and a log-normal distribution for clutch size, and β-distributions for juvenile survival using means and standard deviations based on Biek et al. [43].

Given the dramatic effects of *Bd* on cricket frog survival observed in this study and that not all cricket frogs within a population are infected with *Bd*, we repeated the above analysis to account for variation in *Bd* infection prevalence. We reduced the prevalence of *Bd* from 100% in the original model to values between 10–75% to create a range of infection-prevalences for cricket frog populations in the field. All analyses and modeling exercises were completed in R version 3.6.1.

## Results

Larval exposure to trematodes significantly decreased the number of cricket frog tadpoles that reached metamorphosis by 30% ($\chi^2_{[1,8]}$ = 8.08, p = 0.004; Fig 2A). Trematode exposure significantly reduced mass at metamorphosis by 23% and significantly increased time to metamorphosis by 8% (Table 2; Fig 2B).

qPCR analysis of *Bd* swabs revealed all individuals (100% infection rate) exposed to *Bd* had the presence of *Bd* on their skin 4 weeks post *Bd* exposure. *Bd* was not detected in qPCR analysis of pooled control samples. While individuals that survived overwintering maintained their *Bd* infection status (100% infection rate following overwintering), trematode metacercarial cysts were absent in all animals that survived through overwintering.

Exposure to either trematodes or *Bd* reduced growth in the terrestrial environment (Table 2). As a result, the smallest individuals before overwintering were those exposed to both parasites and the largest individuals were exposed to neither parasite (Fig 3). Cricket frogs exposed to *Bd* were statistically indistinguishable from unexposed controls initially, but over time their growth slowed, whereas frogs exposed to trematodes maintained their smaller size throughout terrestrial rearing (Fig 3).

Although neither infectious agent, combined or individually, influenced terrestrial survival prior to overwintering ($\chi^2_{[1,64]}$ < 0.0014, p ≥ 0.970), exposure to *Bd* had severe impacts on cricket frog overwintering survival. *Bd* infection significantly reduced overwintering survival by 88% ($\chi^2_{[1,62]}$ = 22.41, p < 0.001; Fig 4A); prior exposure to trematodes during larval development did not alter the effect of *Bd* ($\chi^2_{[1,62]}$ = 1.22, p = 0.272) or individually influence overwintering survival ($\chi^2_{[1,62]}$ = 0.11, p = 0.746). The sublethal effects of both parasites on growth in the terrestrial environment also influenced their survival through overwintering, such that smaller individuals were less likely to survive through overwintering ($\chi^2_{[1,64]}$ = 5.99, p = 0.014;

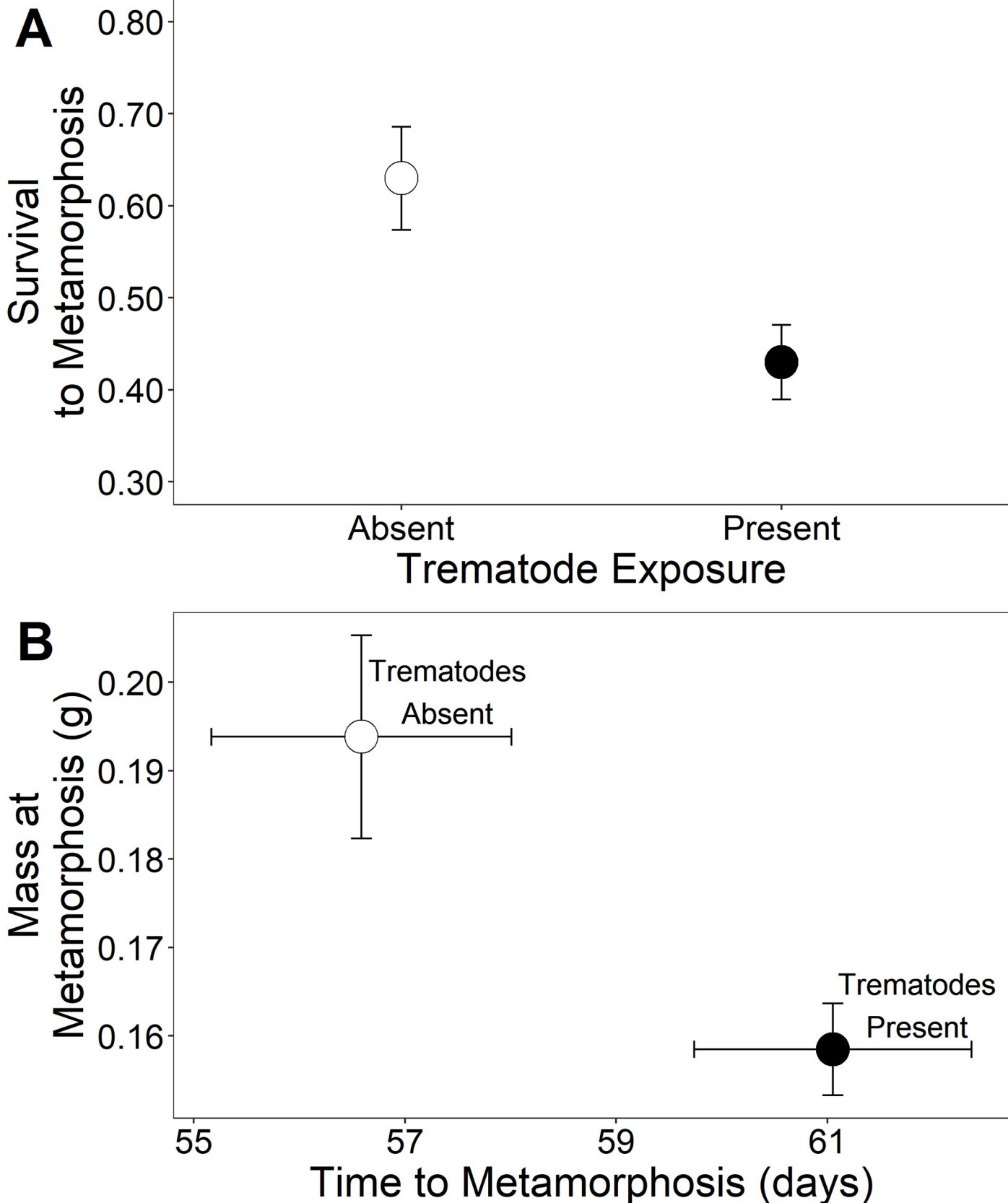

**Fig 2. Effects of trematodes on metamorphic responses.** A) Proportion of cricket frog tadpoles that survived to metamorphosis exposed to the absence or presence of trematodes during larval development. B) Time to metamorphosis and mass at metamorphosis for cricket frogs exposed to the presence or absence of trematode during larval development. Plotted values are means ± 1 SE.

Fig 4B). However, the path analysis revealed that indirect effects of trematodes and *Bd* did not solely explain reductions in survival through overwintering (Fig 5A; Fisher's C = 24.60, p = 0, AICc = 40.95). The addition of a direct path between *Bd* infection resulted in a better fitting model (Fisher's C = 0.09, p = 0.956, AICc = 16.03), as indicated by a p > 0.05 (Fig 5B).

We found that both trematode exposure during larval development and *Bd* infection following metamorphosis had considerable effects on population growth estimates for Blanchard's cricket frogs (Fig 6). Our control model included no exposure to trematodes or *Bd* and had a mean λ of 2.02, indicating rapid population growth. When survival to metamorphosis was decreased by 30%, as observed here with trematode exposure, mean λ decreased to 1.41, indicating the population would still increase rapidly. However, when overwintering survival decreased by 88%, as with *Bd* infections, mean λ reduced to 0.24, resulting in a declining population. Including both trematode and *Bd* exposure in the model, mean λ further declines to 0.17 (Fig 6A). However, when *Bd* prevalence is less than 50%, cricket frog populations are expected to persist, as mean λ remains above 1.00 until *Bd* prevalence is over 50% (Fig 6B).

## Discussion

Species interactions, particularly those that occur between infectious agents and their hosts, play a fundamental role in shaping communities and ecosystem biodiversity. Population-level changes have been attributed to parasitic infections in many wildlife taxa [49] from primates [50] to honeybees [51]. The rise of emerging infectious diseases resulting from redistribution of species around the globe through the animal trade and accidental release poses an additional environmental challenge to species already managing natural parasite and pathogen infections [3, 4]. The amphibian chytrid fungus, *Bd*, has caused global declines, yet in many places it appears to be present in populations without having obvious impacts on population dynamics. As a global exotic pathogen, *Bd* has been particularly devastating in that it impacts thousands of species—a characteristic that makes it fairly unique among pathogens, as most pathogens are specialists, but one that may become more common as emerging infectious diseases spread globally due to human actions [52]. Understanding the range of effects that disease-causing agents can have on species in relevant ecological contexts is important to anticipate responses in natural environments where competition, predation, and other parasites or pathogens can

**Table 2. Summary of ANOVAs for time to and mass at metamorphosis, and repeated measure ANOVA for juvenile mass through time.** Significant effects (α ≤ 0.05) are in bold text.

| Response Variable | Source of variation | df | F value | p-value |
|---|---|---|---|---|
| Time to metamorphosis | Trematodes | 1, 8 | 5.32 | **0.050** |
| Mass at metamorphosis | Trematodes | 1, 8 | 7.85 | **0.023** |
| Terrestrial Growth | | | | |
| Between subjects | Trematodes | 1,62 | 14.59 | **<0.001** |
| | *Bd* | 1, 62 | 13.64 | **<0.001** |
| | Trematodes x *Bd* | 1, 62 | 1.34 | 0.252 |
| Within subjects | Time | 5, 310 | 1565.04 | **<0.001** |
| | Time x Trematodes | 5, 310 | 1.45 | 0.206 |
| | Time x *Bd* | 5, 310 | 11.78 | **<0.001** |
| | Time x Trematodes x *Bd* | 5, 310 | 0.15 | 0.980 |

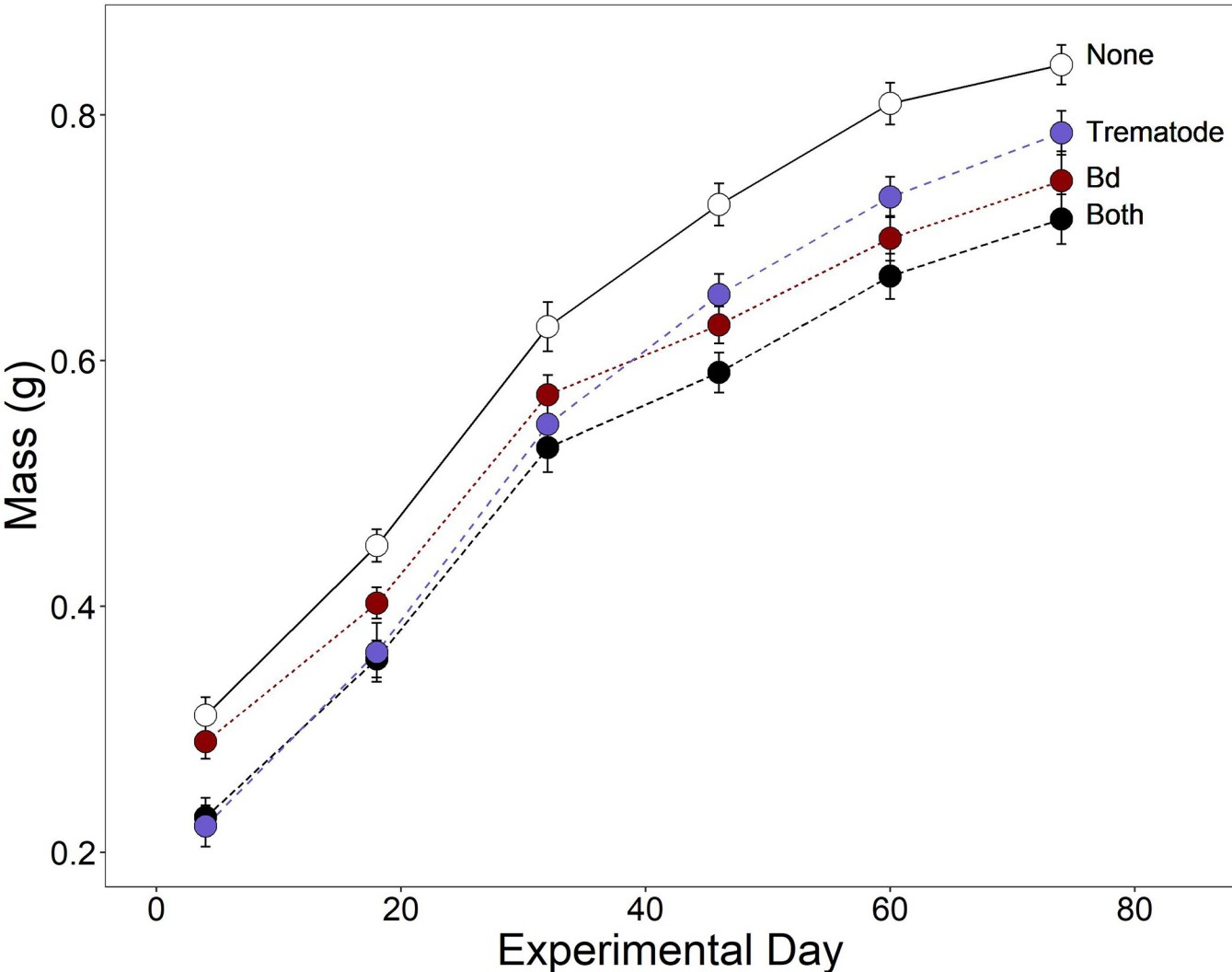

**Fig 3. Growth of cricket frog metamorphs during terrestrial rearing.** Mass of cricket frog metamorphs over time in the terrestrial environment according to treatments (None [for no trematode or Bd exposure], trematodes only, *Bd* only, both infectious disease agents). Plotted values are means ± 1 SE.

influence susceptibility to disease. Indeed, evaluating novel disease agents in relevant ecological contexts is essential to examine how they can change from sublethal to lethal as the conditions shift. Our study demonstrated that *Bd* can negatively impact host size, a trait correlated with fitness, similarly to natural trematode parasites, but that the consequences of *Bd* become more severe than natural parasites when individuals overwinter. Reductions in survival through overwintering could explain enigmatic declines for species where mass mortality has not been observed and suggests that populations may rise and fall depending upon factors that influence parasite prevalence.

### Trematodes impacted larval cricket frogs and set the stage for differential impacts of *Bd*

Trematodes are widespread parasites of amphibians and other taxa with the potential to impact population dynamics through negative effects on individual fitness. Exposure to

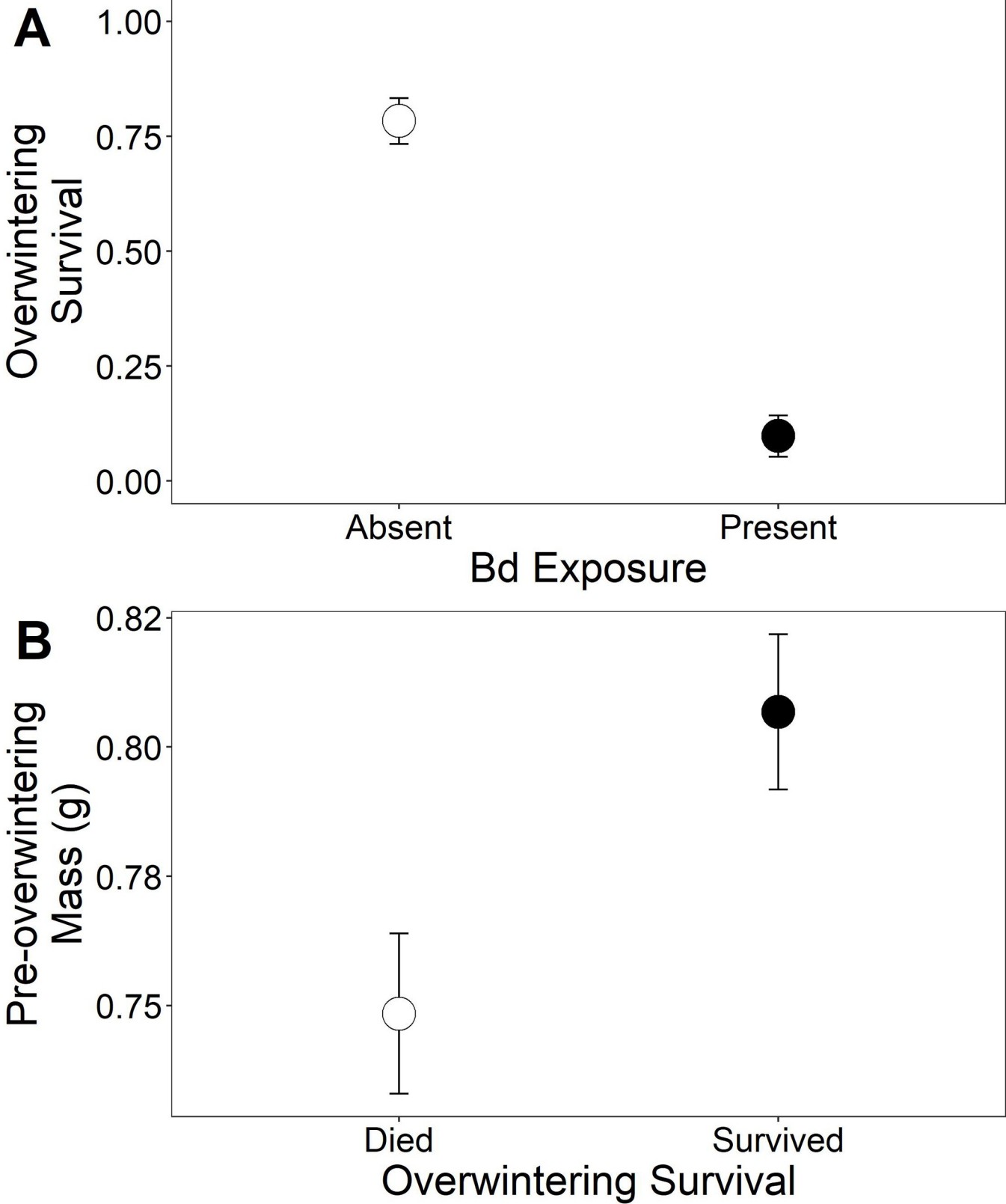

**Fig 4. Effects of *Bd* infection and mass on overwintering survival.** A) Proportion of cricket frog metamorphs exposed to the absence or presence of *Bd* that survived terrestrial overwintering. B) The mass of cricket frogs prior to overwintering based on whether they died or survived overwintering. Plotted values are means ± 1 SE.

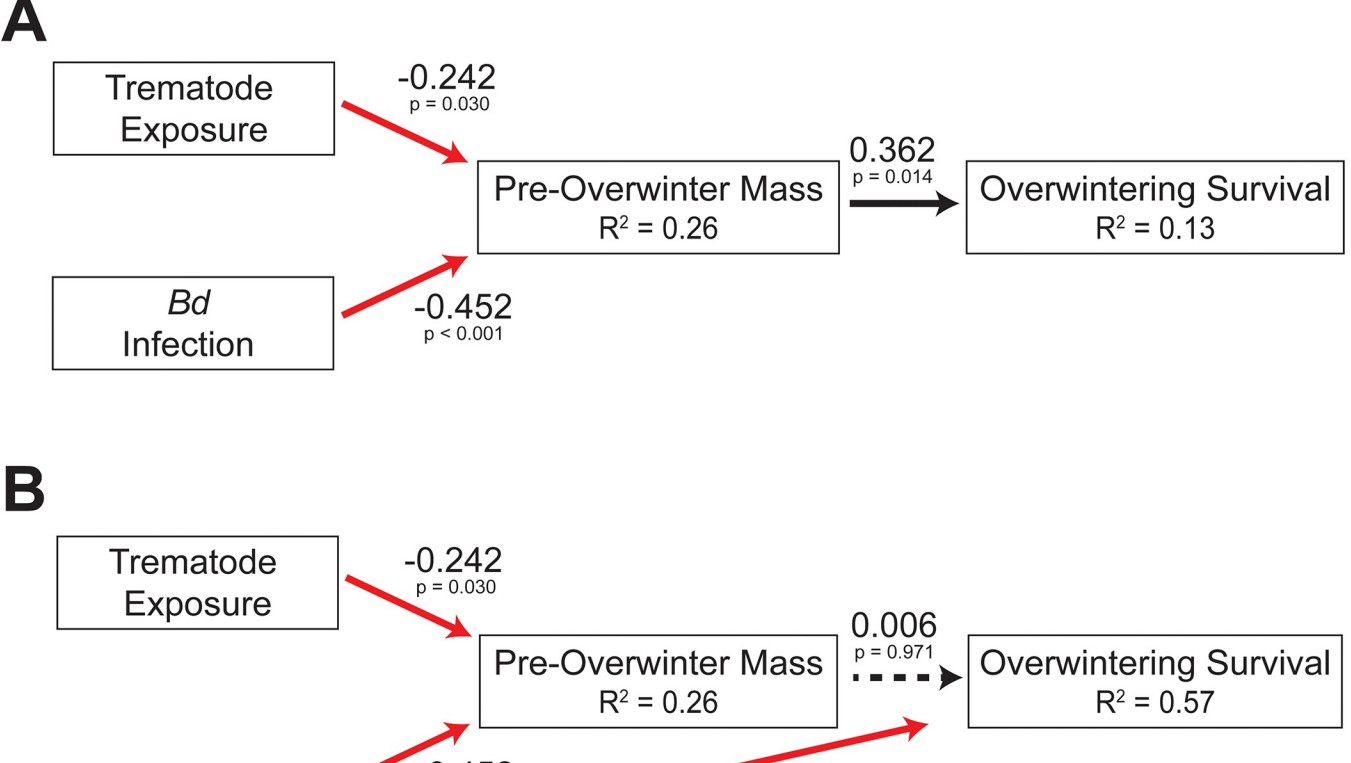

**Fig 5. Proposed relationships between infectious agent exposure and overwintering survival.** A) Piecewise SEM (path analysis) showing the indirect effect of trematode exposure and *Bd* infection on survival through overwintering (Fisher's C = 24.60, p = 0). B) Piecewise SEM after the addition of a missing path between *Bd* infection and survival through overwintering (Fisher's C = 0.09, p = 0.956). Black and red arrows indicate positive and negative relationships, respectively. Solid arrows indicate significant relationships. Standardized path coefficients and p-values are given by numbers near each arrow. $R^2$ values give the sum of the variance explained by all causal paths on a variable.

natural parasites could alter the impact of *Bd* on amphibians, and larval exposure to trematodes had direct negative effects on cricket frogs reared from hatching through metamorphosis in our study (similar to [25, 26]). We found that trematode exposure significantly reduced survival, increased time to metamorphosis, and reduced size at metamorphosis—despite longer larval periods and reduced density as a result of decreased survival. As a species with a small clutch size and annual life cycle, cricket frogs are particularly vulnerable to conditions that reduce metamorphic success or remove individuals from the population prior to breeding each year [41, 42]. Reductions in metamorph survival have a disproportionate impact on amphibian population growth [43], suggesting that trematode exposure could reduce cricket frog population viability. Nevertheless, when we estimated the effect of trematode exposure on cricket frog population viability by reducing survival to metamorphosis in our model, we found that although lambda decreased with trematode exposure, populations were still expected to grow. Our population model, however, did not consider the sublethal effects (i.e., reduced growth) of trematodes on cricket frog fitness, which could also reduce population viability as suggested by the path analysis.

Decreases in mass at metamorphosis can reduce amphibian terrestrial survival and growth [46, 53, 54] leading to reduced fecundity [55]. This may be particularly consequential for

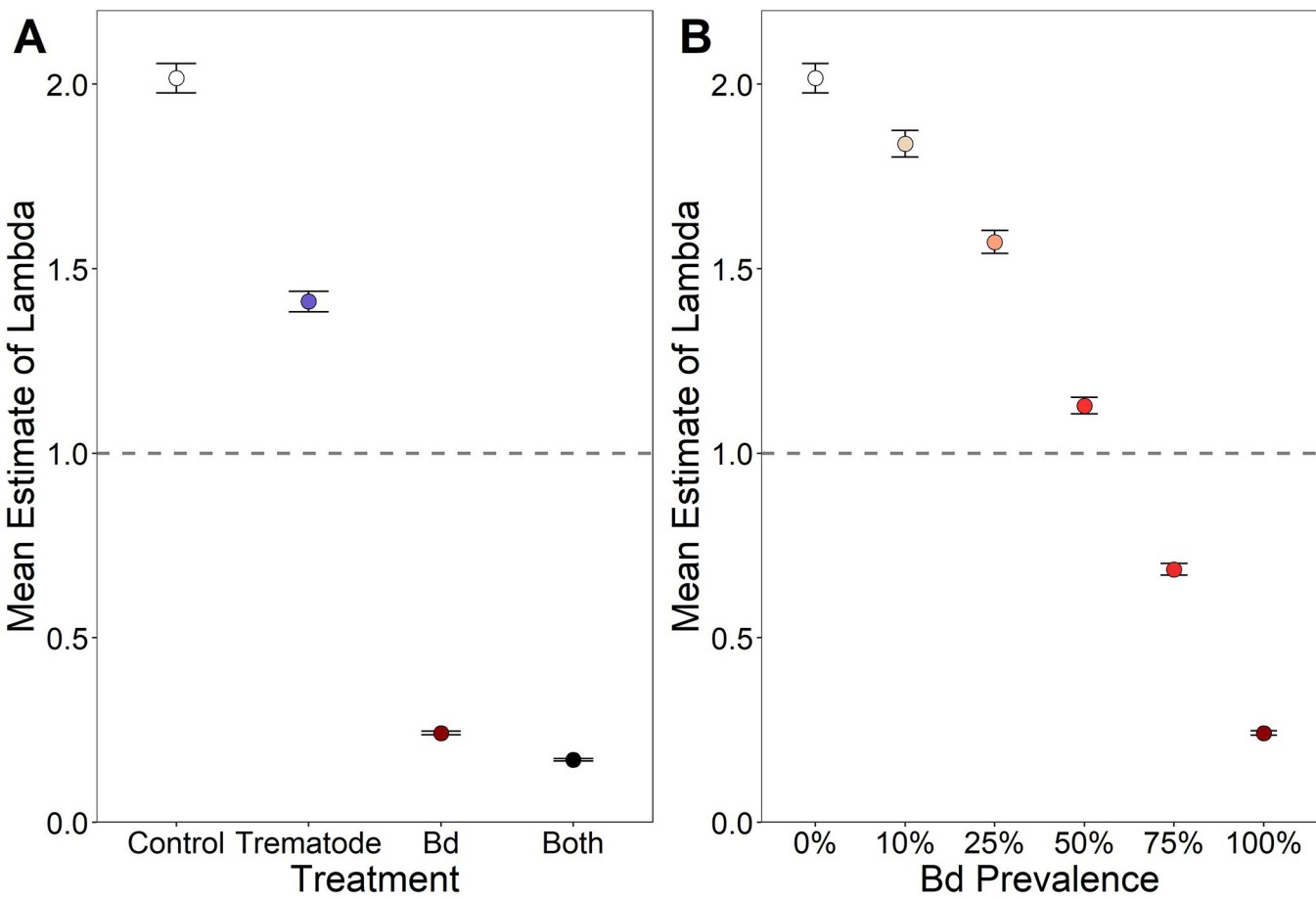

**Fig 6. Effects of parasite treatment and *Bd* prevalence of population growth rates.** A) Mean λ (the finite rate of increase of population growth) values with 95% confidence intervals under experimental conditions. B) Mean λ (the finite rate of increase of population growth) values with 95% confidence intervals determined based on *Bd* infection prevalence in the population.

species that are not able to compensate for reduced size at metamorphosis in the terrestrial environment, as observed in this study. Here, cricket frogs exposed to trematodes maintained their small body size throughout terrestrial rearing and smaller individuals were less likely to survive overwintering. Interestingly, we did not observe trematode cysts within any animal that survived to the end of the experiment, suggesting that cricket frogs may have cleared their trematode infections (as in [56]); because we used all individuals that metamorphosed from trematode ponds in our terrestrial study, only one individual that died during metamorphosis was assessed for and found to have trematode infection at metamorphosis, suggesting that tadpoles were infected with trematodes. The energetic cost associated with trematode clearance may have prevented cricket frogs from allocating energy towards growth [57], thereby reducing their fitness in the terrestrial environment.

### Effects of *Bd* on cricket frogs were sublethal before overwintering and not affected by prior trematode exposure

Trematode exposure could have increased the susceptibility of cricket frogs to *Bd*, yet this did not appear to be the case. Potentially, cross-reactive immunity could have protected trematode-infected individuals, as observed in other amphibian coinfection systems [58]. When

cricket frogs were exposed to *Bd* at metamorphosis, individuals coming from trematode ponds were smaller than from ponds without trematodes; yet exposure to trematodes did not alter the effect of *Bd* or *Bd* infection prevalence. However, *Bd* did have important effects on terrestrial growth, regardless of trematode exposure. Individuals infected with *Bd* at metamorphosis experienced reduced terrestrial growth, an effect that was heightened over time and likely a result of disease progression [19, 21]. Reduced growth can leave individuals more vulnerable to predation [59], less fecund [60, 61], or less likely to reproduce [62, 63]. Yet despite an indication that *Bd* was affecting cricket frogs, it did not impact survival prior to overwintering.

### *Bd*-exposure decimated overwintering cricket frogs

Overwintering is a physiologically demanding period [12, 13] that can influence immune function [64], and it is largely understudied because it is challenging to observe. We found that *Bd* reduced survival during overwintering by 88%, and that effect could be particularly important at the population level, as this degree of mortality in the field could potentially devastate juvenile recruitment [1, 4, 9, 17] and require rescue by frogs dispersing from adjacent populations. Given that *Bd*-infected cricket frogs were smaller at overwintering than frogs from trematode-only or control conditions, size at metamorphosis could have contributed to reduced overwintering survival with *Bd*-infection relative to trematode-only and control frogs. However, similar to our study, Rumschlag and Boone [15] found that northern leopard frogs (*Rana pipiens*) exposed to *Bd* had low overwintering survival in the absence of size differences prior to overwintering. Combined, these results suggest that overwintering may be a critical bottleneck for temperate amphibians that are infected with *Bd* and suggest that infection could lead to enigmatic population declines.

Indeed, when we estimated the effect *Bd* infection on cricket frog population viability by estimating lambda under experimental conditions (i.e., 100% of individuals are exposed to the disease-causing agent(s) in the larval or terrestrial life stages) and found that that if 100% of individuals in a population were exposed to *Bd*, populations would not be able to persist. Population growth rate, however, was sensitive to the proportion of the population that was infected with *Bd;* when *Bd* prevalence exceeded 50%, our model predicted population declines, but not when *Bd* infection was less prevalent. Observed wild cricket frog populations are not far from this threshold with *Bd* prevalence in some populations reaching over 40% [65], which indicates that factors that influence prevalence could have profound impacts on whether a population persists or declines.

## Conclusions

The emergence of novel infectious diseases that successfully spread across the globe bring many potential consequences and in the case of the amphibian chytrid fungus, those consequences have been severe. There has been some relief that many amphibian populations appear to persist in the presence of *Bd*, but also concern that the effects may not be fully apparent given that natural populations deal with multiple stressors. We found that infections of *Bd* or trematodes can impact cricket frog survival and development, and that the co-infection of trematodes and *Bd* can severely limit anuran growth. This study demonstrated that individual infection with trematodes or *Bd* can have sublethal and lethal effects on individuals, particularly during metamorphosis and overwintering. Most notably, we found that individuals infected with *Bd*, regardless of trematode exposure status, were significantly more likely to die during overwintering than uninfected individuals. Because *Bd* is widespread on the landscape and has been detected in populations of Blanchard's cricket frogs for over a decade [30], reduced overwintering survival at the levels we observed in individuals exposed to *Bd* can be a

potential explanation for cricket frog declines and may be important for other species (e.g., northern leopard frogs, see [15]). Yet, the likelihood of local extinction may depend on infection prevalence, which may vary across space and time from differences in environmental conditions [66]. Our study highlights that *Bd* alone can have devastating impacts on survival when the long-term consequences of exposure are examined across relevant life events and that the persistence of populations, therefore, depends upon factors that influence the proportion of individuals infected.

## Acknowledgments

We would like to thank all members of the Boone Amphibian Conservation lab, particularly C. Dvorsky and M. Murphy, for their valuable insights throughout this project, M. Gonzalez, C. Williamson, and E. Overholt for use of microscopy equipment, J. Fruth for assistance at the ERC, M. H. H. Stevens for advice on population modeling, and A. Kiss & X. Deng of the Center for Bioinformatics & Functional Genomics (CBFG) at Miami University for instrumentation support. Work was conducted under Miami University IACUC Protocol #827, and animals were collected under Ohio Division of Wildlife permit #20–177.

## Author Contributions

**Conceptualization:** Olivia Wetsch, Miranda Strasburg, Jessica McQuigg, Michelle D. Boone.

**Data curation:** Olivia Wetsch, Miranda Strasburg, Jessica McQuigg, Michelle D. Boone.

**Formal analysis:** Miranda Strasburg.

**Funding acquisition:** Olivia Wetsch, Miranda Strasburg, Jessica McQuigg, Michelle D. Boone.

**Investigation:** Olivia Wetsch, Miranda Strasburg, Jessica McQuigg.

**Methodology:** Olivia Wetsch, Miranda Strasburg, Jessica McQuigg, Michelle D. Boone.

**Project administration:** Olivia Wetsch, Miranda Strasburg, Jessica McQuigg, Michelle D. Boone.

**Resources:** Olivia Wetsch, Miranda Strasburg, Jessica McQuigg, Michelle D. Boone.

**Software:** Olivia Wetsch, Miranda Strasburg, Jessica McQuigg, Michelle D. Boone.

**Supervision:** Olivia Wetsch, Miranda Strasburg, Jessica McQuigg, Michelle D. Boone.

**Validation:** Olivia Wetsch, Miranda Strasburg, Jessica McQuigg, Michelle D. Boone.

**Visualization:** Olivia Wetsch, Miranda Strasburg, Jessica McQuigg, Michelle D. Boone.

**Writing – original draft:** Olivia Wetsch.

**Writing – review & editing:** Olivia Wetsch, Miranda Strasburg, Jessica McQuigg, Michelle D. Boone.

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
