## [Decision Letter · Decision Letter 0]

22 Nov 2021

PONE-D-21-30796Is overwintering mortality driving enigmatic declines? Evaluating the impacts of trematodes and the amphibian chytrid fungus on an anuran from hatching through overwinteringPLOS ONE

Dear Authors,

Thank you for submitting your manuscript to PLOS ONE. After careful consideration, we feel that it has merit but does not fully meet PLOS ONE’s publication criteria as it currently stands. Therefore, we invite you to submit a revised version of the manuscript that addresses the points raised during the review process.

<ul><li>I apologize for the delay in returning reviews of your manuscript.  It was difficult to find reviewers although I solicited reviews from quite a few experts. Your manuscript has now been reviewed by two experts on amphibian diseases and chytridiomycosis.  In general, both were positive about the content of the paper. However, each reviewer made a number of suggestions that would greatly improve the manuscript. Please revised and submit for further review.  Please carefully consider all of the reviewer comments and provide a point-by-point response.

Reviewer 1 was concerned about the statistical treatment of the data.  There is a need for a more complete analysis about the effect of *Bd *loads.  This reviewer made detailed suggestions about how to re-work the statistical analysis.  Reviewer 1 also made a number of suggestions about the presentation of the findings.  Instead of *Bd* exposure, it is likely that you were looking at *Bd* infection, and including the *Bd* loads in the analysis is important.Reviewer 2 would like you to consider and explain more clearly the rationale for the experimental design.

We look forward to receiving your revised manuscript.

Kind regards,

Louise A. Rollins-Smith

Academic Editor

PLOS ONE

2. In your Methods section, please provide additional location information, including geographic coordinates of your field collection site if available.

4. We note you have included a table to which you do not refer in the text of your manuscript. Please ensure that you refer to Table 1 in your text; if accepted, production will need this reference to link the reader to the Table.

Reviewers' comments:

Reviewer's Responses to Questions

**Comments to the Author**

1. Is the manuscript technically sound, and do the data support the conclusions?

Reviewer #1: Yes

Reviewer #2: Yes

2. Has the statistical analysis been performed appropriately and rigorously? 

Reviewer #1: No

Reviewer #2: Yes

3. Have the authors made all data underlying the findings in their manuscript fully available?

Reviewer #1: Yes

Reviewer #2: Yes

4. Is the manuscript presented in an intelligible fashion and written in standard English?

Reviewer #1: Yes

Reviewer #2: Yes

5. Review Comments to the Author

Reviewer #1: PONE-D-21-30796_review

In this paper the authors looked at the impact of two parasites on mortality and size outcomes in the cricket frog. Additionally, they looked at the impacts these pathogens have on overwintering success. The work is strong and I think the paper could use a little reworking (including running some new stats) to strengthen the presentation of this very cool and important study.

The Short title might be more accurate to change parasite for Pathogen.

Line 28- Chytrid is not just an amphibian fungus, please consider just saying novel pathogenic chytrid fungus.

Line 30- early life experience of what?

Line 31-32- Can you rearrange this statement. As it stands the negative impact of trematodes on metamorphosis is almost like a 2nd thought, when it’s a valid finding. I’d say something like prior exposure to a trematode negatively impacted metamorphosis but did not influence the effect of Bd-exposure and that Bd-exposure alone resulted in…. (use and not but because the stated results are in addition to what you just described not in conflict)

Line 32- Are you sure it’s just Bd-exposure and not Bd infection? Most of the amphibians in the study were likely exposed to Bd.

Line 46- would co-infections work here from a terminology perspective?

Line 50- overwintering is not really a part of their life cycle, but is more a part of their natural history

Line 56-58- I think it would help with flow if you bring up Bd and trematodes prior to introducing them separately in their own paragraphs. Maybe it can just be added to the statement that is there, with a statement like- “Given that disease-driven mortality may be difficult to detect during metamorphosis and overwintering when mortality is not easily observed in the field, it is critical to evaluate the impacts of commonly co-occurring pathogens, like Bd and trematodes, across multiple life stages and through critical life events to better understand enigmatic declines.”

Line 68- Can you make it a little more clear what you mean by “certain conditions”?

Line 71-72- can you make it clear that the changes in the trematode abundance due is to the increase in the intermediate host influenced by nutrient inputs

Line 75-76- I think it’s worth just putting in the parasite names here in the order the exposure is happening. You exposed them to trematodes + then Bd, but not the other way around (I think at least based on your abstract)

Line 129- That’s a heck of a range. Is there an average for this?

Line 142-143- move this summary up to the beginning of the paragraph

Line 160- How did you dose the frogs with Bd? Dorsal exposure? Submerged in water?

Line 168- Spin baskets sounds cool but I don’t know what it is, could you add a bit more here

Line 173- James et al? or just James?

Line 188- Any chance you looked at (or could look at) Fat bodies? It’s not necessary but would be an incredibly strong indicator of health

Line 196- Again I think you were really looking at trematode and Bd infection not exposure, correct? You have the actual Bd data.

Line 247- What about actual trematode abundance? Does the actual infection load correlate with these factors or just exposure? I think you need to include all the actual infection load numbers and information because you took that data! For example can you compare actual load for trematodes (and Bd when applicable) to the actual mass or growth rate?

Line 282- Can you use a path analysis (e.g. laavan package in R) to see if there’s an indirect effect of trematodes on mortality through overwintering. You describe a trematodes� size � overwintering effect.

Lines 284-286-The idea that all individuals are positives strengthens my argument earlier that you should be running all of these stats with the actual Bd Load not just “exposed” individuals. It’s not bad to do Bd+ vs Bd- stats but you have way more information you can and should explore here with the actual quantitative data

Line 313-324- Combine sentences 1 + 2 here, the first sentence is really too broad for this article and tighten up this first paragraph too. You’ve already set the scene for the work here so it’s not really worth going back to REALLY broad ecology. Shrink this down to one strong sentence.

Lines 328-331- To me, this sentence doesn’t resonate to with what you resented in the results. What traits did Bd negatively impact? Maybe you mean size? Can you rework this sentence so that it is clear and specific to your findings? Then tie this in to the sentence starting on line 331 more specifically.

Lines 342-344- unpack this a little more, I’m not positive I’m totally following what you mean here. So- I see the justification in the next sentence maybe flop these sentences so the reader is with you as you make the viability statement

Line 349- there are also papers showing reduced size = increased depredation risk too

Line 352-354- This fact should be in the results not just in the discussion. If I missed it, sorry but I don’t remember seeing it. Did the ones that died early have trematodes? Were you able to check? Bd Loads should also be reported in the results too

Line 355-356- look up some of the energy budget papers (many are in the insect models but the take home is the same). There’s a limited energy budget that can be utilized (worth citing a paper referencing this here)

Line 372- Important findings!

Line 385- This paragraph has a feeling of being added on at the end. The model is interesting and I think it would be better to try to integrate it into the discussion a little more rather than just having it be a single separate paragraph. When you present findings earlier in the discussion reference whether your model supports those findings or not.

Figure 3- looks like there are synergistic effects of both parasites. Can you run the stats to see if that’s true and really highlight that if it is.

Reviewer #2: Is overwintering mortality driving enigmatic declines? Evaluating the impacts of

trematodes and the amphibian chytrid fungus on an anuran from hatching through

overwintering

In this manuscript, the authors aim to determine the interactive effects of Bd and trematodes on cricket frogs. The study is an important addition to the field and the experiment is well-executed. The primary critique I have with the manuscript concerns some of the rationale for the experiment and the way(s) in which amphibian declines are discussed. With minor revisions, however, I believe the manuscript will be suitable for publication.

Please see specific comments below.

Introduction

Line 45: “have a limited understanding” – the authors do this throughout the manuscript. They mention that little or limited is known about thing X, Y, or Z. What classifies as “little” or “limited”? One study? 10? 100? Reiterating from above, the manuscript deserves publication. However, I encourage the authors to move beyond the “we don’t know much about this” rationale to find more robust reasons for study.

Line 48: Define “concerning.”

Lines 60-62: For those unfamiliar with Bd, be sure to mention that you’re talking about effects to amphibians specifically.

Line 70-72: Please emphasize the relevance of this statement.

Methods

Line 110: Explain the developmental stages of trematodes.

Discussion

Lines 314-315: Is this actually true, i.e., “very common”?

Line 323: CITE “fairly unique.”

Lines 359-360: Why do the authors think this counter result occurred?

Line 385: Are the declines still poorly understood?

6. PLOS authors have the option to publish the peer review history of their article (what does this mean?). If published, this will include your full peer review and any attached files.

Reviewer #1: No

Reviewer #2: No

---

## [Author Response · Author response to Decision Letter 0]

21 Dec 2021

PONE-D-21-30796

Is overwintering mortality driving enigmatic declines? Evaluating the impacts of trematodes and the amphibian chytrid fungus on an anuran from hatching through overwintering

PLOS ONE

Dear Authors,

Thank you for submitting your manuscript to PLOS ONE. After careful consideration, we feel that it has merit but does not fully meet PLOS ONE’s publication criteria as it currently stands. Therefore, we invite you to submit a revised version of the manuscript that addresses the points raised during the review process.

• I apologize for the delay in returning reviews of your manuscript. It was difficult to find reviewers although I solicited reviews from quite a few experts. Your manuscript has now been reviewed by two experts on amphibian diseases and chytridiomycosis. In general, both were positive about the content of the paper. However, each reviewer made a number of suggestions that would greatly improve the manuscript. Please revised and submit for further review. Please carefully consider all of the reviewer comments and provide a point-by-point response.

We have addressed the reviewer’s comment and believe they have greatly improved our manuscript. 

• Reviewer 1 was concerned about the statistical treatment of the data. There is a need for a more complete analysis about the effect of Bd loads. This reviewer made detailed suggestions about how to re-work the statistical analysis. 

• Reviewer 1 also made a number of suggestions about the presentation of the findings. Instead of Bd exposure, it is likely that you were looking at Bd infection, and including the Bd loads in the analysis is important.

• Reviewer 2 would like you to consider and explain more clearly the rationale for the experimental design.

General Editor Feedback

 #1 We made these changes throughout. 

2. In your Methods section, please provide additional location information, including geographic coordinates of your field collection site if available.

 #2 We added this information (see lines 113, 120, 122).

 #3 We corrected this. 

4. We note you have included a table to which you do not refer in the text of your manuscript. Please ensure that you refer to Table 1 in your text; if accepted, production will need this reference to link the reader to the Table.

 #4 We made this change (see line 267). 

Reviewer #1

In this paper the authors looked at the impact of two parasites on mortality and size outcomes in the cricket frog. Additionally, they looked at the impacts these pathogens have on overwintering success. The work is strong and I think the paper could use a little reworking (including running some new stats) to strengthen the presentation of this very cool and important study.

The Short title might be more accurate to change parasite for Pathogen.

#5 We referred to Bd as a parasite throughout the manuscript, which is common in the literature (see Bielby et al., 2015 Scientific Report; Greenspan et al., 2018 Scientific Reports; Saurer et al., 2020 Ecology) including manuscripts published in this journal (see Gabor et al., 2015, PLoS One). However, we changed all references to Bd as “pathogen” throughout, including the short title. 

Line 28- Chytrid is not just an amphibian fungus, please consider just saying novel pathogenic chytrid fungus.

#6 We made this change (see line 29). 

Line 30- early life experience of what?

#7 We clarified this (see lines 30-32).

Line 31-32- Can you rearrange this statement. As it stands the negative impact of trematodes on metamorphosis is almost like a 2nd thought, when it’s a valid finding. I’d say something like prior exposure to a trematode negatively impacted metamorphosis but did not influence the effect of Bd-exposure and that Bd-exposure alone resulted in…. (use and not but because the stated results are in addition to what you just described not in conflict)

#8 We made this change (see lines 32-34). 

Line 32- Are you sure it’s just Bd-exposure and not Bd infection? Most of the amphibians in the study were likely exposed to Bd.

#9 We change to Bd-infection throughout the manuscript. We did confirm Bd-presence in exposed animals and Bd absence in unexposed animals. We have no reason to believe that Bd control animals were ever exposed to Bd. All animals used in this experiment were reared in artificial pond communities free of Bd, and we were very careful not to contaminate Bd-control individuals during the terrestrial rearing portion of the experiment. Indeed, as mentioned previously, we confirmed that Bd infection was absent in Bd control animals. 

Line 46- would co-infections work here from a terminology perspective?

#10 We made this change (see line 48). 

Line 50- overwintering is not really a part of their life cycle, but is more a part of their natural history

#11 We made this change (see line 53-54). 

Line 56-58- I think it would help with flow if you bring up Bd and trematodes prior to introducing them separately in their own paragraphs. Maybe it can just be added to the statement that is there, with a statement like- “Given that disease-driven mortality may be difficult to detect during metamorphosis and overwintering when mortality is not easily observed in the field, it is critical to evaluate the impacts of commonly co-occurring pathogens, like Bd and trematodes, across multiple life stages and through critical life events to better understand enigmatic declines.”

 #12. We made this change (see lines 61-64). 

Line 68- Can you make it a little more clear what you mean by “certain conditions”?

#13 We clarified this (see lines 74-75). 

Line 71-72- can you make it clear that the changes in the trematode abundance due is to the increase in the intermediate host influenced by nutrient inputs

#14. We clarified this (see line 78). 

Line 75-76- I think it’s worth just putting in the parasite names here in the order the exposure is happening. You exposed them to trematodes + then Bd, but not the other way around (I think at least based on your abstract)

#15. We added this (see lines 86-87). 

Line 129- That’s a heck of a range. Is there an average for this?

 #16. We add in the average snail density (see line 155). 

Line 142-143- move this summary up to the beginning of the paragraph

#17. We made this change (see lines 135-140). 

Line 160- How did you dose the frogs with Bd? Dorsal exposure? Submerged in water?

#18 We clarified this (see lines 187-189). 

Line 168- Spin baskets sounds cool but I don’t know what it is, could you add a bit more here

#19 We clarified this (see lines 201-205). 

Line 173- James et al? or just James?

#20. James was the sole author of this paper. 

Line 188- Any chance you looked at (or could look at) Fat bodies? It’s not necessary but would be an incredibly strong indicator of health

#21. We did not look at fat bodies, but we agree that this would have been a good addition to the study. 

Line 196- Again I think you were really looking at trematode and Bd infection not exposure, correct? You have the actual Bd data.

#22. We made this change throughout. We confirmed Bd infection with qPCR, and agree that claiming Bd infection is accurate, but because after overwintering, metacercariae were absent from the surviving frogs exposed to trematodes, despite trematode-associated effects at metamorphosis, we think it is more appropriate use trematode exposure rather than infection. We did not confirm trematode-exposure at metamorphosis in some individuals in this study, but in other studies we’ve conducted in this study we did have high trematode infection rates for all exposed individuals. 

Line 247- What about actual trematode abundance? Does the actual infection load correlate with these factors or just exposure? I think you need to include all the actual infection load numbers and information because you took that data! For example can you compare actual load for trematodes (and Bd when applicable) to the actual mass or growth rate?

#23 As we mentioned in the results and the discussion (see lines 314-315 and 430-435), after overwintering juveniles were not infected with trematodes, so the loads of all individuals post overwinter had an infection load of zero. We could not measure trematode load at metamorphosis, because survival was reduced by trematodes in the larval environment to the extent that we needed to use all of the trematode-exposed metamorphs that came out of the mesocosms in the terrestrial portion of the experiment to meet our terrestrial experimental design goals. We dissected one metamorph that was found dead a trematode infected mesocosm, and it did contain Echinostoma metacercaria (we added this to the manuscript see lines 167-169). Because confirming load/infection requires dissection, we do not know each individual’s trematode load at the start of terrestrial rearing, but we clarified this in the revision.

Line 282- Can you use a path analysis (e.g. laavan package in R) to see if there’s an indirect effect of trematodes on mortality through overwintering. You describe a trematodes� size � overwintering effect.

#24 We added this analysis (see lines 243-250, 336-339, 350-359, and 423-424). 

Lines 284-286-The idea that all individuals are positives strengthens my argument earlier that you should be running all of these stats with the actual Bd Load not just “exposed” individuals. It’s not bad to do Bd+ vs Bd- stats but you have way more information you can and should explore here with the actual quantitative data

 #25 We did use qPCR in attempt to quantify Bd load in individuals, but we experienced environmental inhibition due to the wooden sticks used for swabbing, soil, and other debris that was on the frog’s skin when swabbed. To overcome inhibition in our samples, we diluted extensively (see McKee et al., 2015 Biological Conservation), which allowed us to get qualitative results regarding the presence or absence of Bd infection, but introduced variation between samples and reduced precision from any given individual, thereby reducing our ability to accurately quantify the pathogen load. For this reason, we chose to use our assay for only qualitative confirmation of infection. We have clarified this in the revision (see lines 209-213).

Line 313-324- Combine sentences 1 + 2 here, the first sentence is really too broad for this article and tighten up this first paragraph too. You’ve already set the scene for the work here so it’s not really worth going back to REALLY broad ecology. Shrink this down to one strong sentence.

#26 We made this change (see lines 379-382).

Lines 328-331- To me, this sentence doesn’t resonate to with what you resented in the results. What traits did Bd negatively impact? Maybe you mean size? Can you rework this sentence so that it is clear and specific to your findings? Then tie this in to the sentence starting on line 331 more specifically.

#27 We clarified this (see lines 397-403).

Lines 342-344- unpack this a little more, I’m not positive I’m totally following what you mean here. So- I see the justification in the next sentence maybe flop these sentences so the reader is with you as you make the viability statement

#28 We made this change (see lines 412-419). 

Line 349- there are also papers showing reduced size = increased depredation risk too

 #29 This is true in larval amphibians, but very little is known about how size influences predation risk in terrestrial amphibians, although presumably larger individuals are better able to avoid capture. Because we are not considering predation in this paper, we do not feel adding references about size and predation risk is necessary. 

Line 352-354- This fact should be in the results not just in the discussion. If I missed it, sorry but I don’t remember seeing it. Did the ones that died early have trematodes? Were you able to check? Bd Loads should also be reported in the results too

#30 We have clarified this in the revision. This was in results of the initial manuscript draft (see lines 342-343). See comment #23 for clarification on trematode load. See comment #25 for clarification on Bd load.

Line 355-356- look up some of the energy budget papers (many are in the insect models but the take home is the same). There’s a limited energy budget that can be utilized (worth citing a paper referencing this here)

#31 We added a reference here (see line 436) 

Line 372- Important findings!

Line 385- This paragraph has a feeling of being added on at the end. The model is interesting and I think it would be better to try to integrate it into the discussion a little more rather than just having it be a single separate paragraph. When you present findings earlier in the discussion reference whether your model supports those findings or not.

#32 We integrated this information with the previous paragraphs (see lines 419-422 and 465-474). 

Figure 3- looks like there are synergistic effects of both parasites. Can you run the stats to see if that’s true and really highlight that if it is.

#33 We tested for an interaction between Bd and trematodes on terrestrial growth in our initial manuscript (see Table 2). There was not a significant interaction between the two parasites. 

Thank you, Reviewer #1 for the helpful comments! 

Reviewer #2

In this manuscript, the authors aim to determine the interactive effects of Bd and trematodes on cricket frogs. The study is an important addition to the field and the experiment is well-executed. The primary critique I have with the manuscript concerns some of the rationale for the experiment and the way(s) in which amphibian declines are discussed. With minor revisions, however, I believe the manuscript will be suitable for publication.

Please see specific comments below.

Introduction

Line 45: “have a limited understanding” – the authors do this throughout the manuscript. They mention that little or limited is known about thing X, Y, or Z. What classifies as “little” or “limited”? One study? 10? 100? Reiterating from above, the manuscript deserves publication. However, I encourage the authors to move beyond the “we don’t know much about this” rationale to find more robust reasons for study.

#34 We rephrased this sentence (and others throughout) to emphasize that co-infections are common, and therefore deserve attention (see lines 47-50). 

Line 48: Define “concerning.”

#35 We clarified this (see lines 51-52). 

Lines 60-62: For those unfamiliar with Bd, be sure to mention that you’re talking about effects to amphibians specifically.

#36 We clarified this (see lines 65 and 67). 

Line 70-72: Please emphasize the relevance of this statement.

#37 We clarified this (see lines 78-79). 

Methods

Line 110: Explain the developmental stages of trematodes.

#38 We added this information (see lines 122-129). 

Discussion

Lines 314-315: Is this actually true, i.e., “very common”?

#39 We removed this in the revision. 

Line 323: CITE “fairly unique.”

#40 We clarified this and added a citation (see lines 390-391). 

Lines 359-360: Why do the authors think this counter result occurred?

#41 We added an explanation here (see lines 440-441).

Line 385: Are the declines still poorly understood?

#42 This sentence was removed with the revision.

---

## [Editor Report · Decision Letter 1]

30 Dec 2021

Is overwintering mortality driving enigmatic declines? Evaluating the impacts of trematodes and the amphibian chytrid fungus on an anuran from hatching through overwintering

PONE-D-21-30796R1

Dear Authors,

We’re pleased to inform you that your manuscript has been judged scientifically suitable for publication and will be formally accepted for publication once it meets all outstanding technical requirements.

Kind regards,

Louise A. Rollins-Smith

Academic Editor

PLOS ONE

I have looked over your revised manuscript and the response to reviews.  It is my opinion that you have done a good job of addressing the reviewer’s concerns.  Both reviewers consider this to be an important addition to the literature, and so it is my opinion that the manuscript is now acceptable for publication.

---

## [Editor Report · Acceptance letter]

6 Jan 2022

PONE-D-21-30796R1 

Is overwintering mortality driving enigmatic declines? Evaluating the impacts of trematodes and the amphibian chytrid fungus on an anuran from hatching through overwintering 

Dear Dr. Wetsch:

I'm pleased to inform you that your manuscript has been deemed suitable for publication in PLOS ONE. Congratulations! Your manuscript is now with our production department. 

Kind regards, 

on behalf of

Dr. Louise A. Rollins-Smith 

Academic Editor

PLOS ONE